# The Effect of Changes in Settings from Multiple Filling Points to a Single Filling Point of an Industry 4.0-Based Yogurt Filling Machine

**Jinping Chen** [1] , **Razaullah Khan** [2,*] , **Yanmei Cui** [3,*] , **Bashir Salah** [4] , **Yuanpeng Liu** [1] **and Waqas Saleem** [5]

1   School of Aerospace Engineering, Zhengzhou University of Aeronautics, Zhengzhou 450046, China
2   Engineering Management Program, Institute of Manufacturing, University of Engineering and Applied Sciences, Swat 19060, Pakistan
3   School of Mechanical Engineering, Shanghai Dianji University, Shanghai 201306, China
4   Industrial Engineering Department, College of Engineering, King Saud University, P.O. Box 800, Riyadh 11421, Saudi Arabia
5   Department of Mechanical and Manufacturing Engineering, Institute of Technology, F91 YW50 Sligo, Ireland
*   Correspondence: razaullah@ueas.edu.pk (R.K.); hongyong83@126.com (Y.C.)

**Abstract:** In process optimization, a process is adjusted so as to optimize a set of parameters while meeting constraints, with the objective to either minimize the total processing time or maximize the throughput. This article focused on the process optimization of a fully automated yogurt and flavor-filling machine developed based on the industrial revolution 4.0 concept. Mathematical models were developed for minimizing the total processing time or maximizing the throughput of an Industry 4.0-based yogurt filling system with two different machine settings called Case-I and Case-II. In Case-I, the yogurt and flavors are filled at two distinct points while Case-II considers the filling of yogurt and flavors at a single point. The models were tested with real data and the results revealed that Case-II is faster than Case-I in processing a set of customer orders. The results were used as inputs for the single-dimension rules to check which one results in more intended outputs. Additionally, different performance measures were considered and the one with most importance to the management was selected.

**Keywords:** mathematical modeling; production scheduling; process optimization; advanced optimization methodologies; modeling of industrial processes



## 1. Introduction

The Industry 4.0 concept has sparked worldwide attention from various production sectors. In order to conceptualize the idea in the manufacturing sectors, policy makers of different countries initiated their funded mega-research projects to uplift and transform existing technologies according to the modern needs of industry to achieve dynamic customer demands [1–8]. The idea of Industry 4.0 was proposed by Germany as one of the key initiatives towards its high-tech strategy to uplift the manufacturing sector and achieve its desired goals [9]. This policy shift posed a threat to European policies and compelled the leading nations to initiate their own strategies towards developing their own dominant technological manufacturing sectors. For instance, the United States has initiated similar initiatives, known as Smart Manufacturing, to compete in the new technological revolution [10]. In 2014, China proposed the "Made in China 2025" vision to break China's reliance on the foreign technology and transform China into a world-leading manufacturing power [11,12]. In 2016, Japan introduced an advanced variant to Industry 4.0, called Society 5.0, which went far beyond Industry 4.0, whose aim was the digitization of all sectors of the Japanese society [13]. In parallel, Saudi Arabia announced its strategic framework

known as "Vision 2030" to update the existing industries (refinery, petrochemical, fast-moving consumer goods, etc.) within the kingdom according to modern technological revolutions [14].

The basic purpose of the Vision 2030 framework is to renovate, standardize, and digitally transform the Kingdom of Saudi Arabia. A step towards this strategy was taken by the laboratory of Computer Integrated Manufacturing (CIM) at the Department of Industrial Engineering, King Saud University, Saudi Arabia by designing an automatic yogurt-filling system for automatic filling of yogurt and addition of different flavors as per customers' demands [15]. This approach has produced consistent outcomes and studies have been published in peer-reviewed journals. For example, in Virtual Reality (VR)-based engineering education, the precision of robotic arms and conveyor belts was shown to improve production sustainability in Industry 4.0 [16]. A thorough description of the yogurt-filling method was also offered in Industry 4.0-based real-time scheduling and dispatching of lean production environments [17].

Industry 4.0 systems involve a variety of agents, from factories to end customers, in which the high-connectivity and cooperation can optimize operational processes, products, and services [18]. B. Salah [19] performed a research study of the yogurt filling system capable of mixing the yogurt with three different flavors as demanded by the customers. The number of containers used in the machine were four. In one container, the base yogurt was stored, while in the remaining three constrainers, three different flavors were stored. The empty cups were automatically moved to a specific location called the bottle feeding point and were then entered into the machine through the entry point over the conveyor belt. Upon reaching the single filling point, the bottles were filled with the required volumes of yogurt and flavors. The feed rate of the nozzles was controlled through solenoid valves. Four diaphragm pumps were used to deliver the yogurt and flavors from the tanks to the unified head of nozzles. The valves with a specific feed rate value were opened for a certain time to fill the bottles with the required volume of yogurt and flavors. The completely filled bottles were then moved to the exit point over the belt where a robotic arm was used to remove the bottles from the system. The whole process was made automatic with increased throughput and less human involvement. Additionally, Node-RED was used to run the system, which has the capability of machine-to-machine communication as it can connect multiple devices. There were several challenges faced during the project and few of them were minimizing the overall cost of the system, programming expertise while managing the Raspberry Pi, and the slow speed of the internet. The design was divided into two phases called the pre- and post-production phases. In the study, only the pre-production phase was considered while the post production phase in which the completely filled bottles are to be stored in the refrigerator storage system will be studied in future research. M. Ramadan et al. [20] created a novel real-time manufacturing cost-monitoring system (RT-MCT) that combines lean manufacturing and RFID ideas. The RT-MCT was designed to connect lean operational characteristics with financial expenses in real time. F. A. German et al. [21] conducted a survey in several manufacturing businesses to investigate the deployment of Industry 4.0 technologies and developed an Industry 4.0 technology layer structure. They demonstrated the levels of adoption of these technologies as well as their implications for manufacturing firms. F. Longo et al. [22] provided a human-centered strategy to improve operators' skills and competencies in the context of the new smart factory. The proposed research activity focused on the design and implementation of a practical solution called Sophos-MS, which is capable of integrating augmented-reality contents and intelligent teaching systems with cutting-edge fruition technologies to assist operators in complicated man–machine interactions. M. E. Leusin et al. [23] suggested a system with self-configuring characteristics to deal with production line disruptions. The performance of the suggested framework was evaluated in a simulation study based on a real-world industrial example. Their findings supported benefits in flexibility, scalability, and efficiency achieved by data sharing between manufacturing levels. D. A. Rossit et al. [24] proposed Smart Scheduling, a novel decision-making schema designed to provide flexible and efficient production

plans on the fly while taking advantage of the peculiarities of these new contexts. P. Chen et al. [25] sought to investigate how environmental collaboration across organizational borders influences green innovation from the standpoint of social capital. The study provided an important contribution to the literature by thoroughly exploring how environmental collaboration in emerging nations affects green innovation from the standpoint of social capital. P. Zawadzki et al. [26] developed a broad notion of smart design and production control as key determinants for efficient and reliable operation of a smart factory. Several strategies were offered to help in the design process of personalized goods and the organization of their manufacturing in the context of realizing the mass customization strategy, allowing for a shorter period of creation for a new product. A. Grassi et al. [27] developed a revolutionary architecture for production planning and control with a semi-hierarchical structure in which various management levels were distinguished by their physical identity as well as their functional scope. The suggested architecture represented an improved application of the Industry 4.0 decentralized decision idea, allowing for a better understanding and management of system performance to yield higher profit and response time predictability in highly customized scenarios. P. Spenhoff et al. [28] established an "every product every cycle" (EPEC) 4.0 production control approach, which intended to schedule the production system as effectively as possible while providing the required flexibility and minimum schedule disturbance. L. E. Quezada et al. [29] provided operational excellence as a means of achieving sustainable development goals via Industry 4.0. C. Santos [30] examined some of the important European Union (EU) industrial standards, roadmaps, and scientific publications that led to the portrayal of the phrase Industry 4.0, as well as how key technologies and concepts have evolved over time. B. Salah et al. [31] presented the second phase design of control architecture for the yogurt filling machine based on Industry 4.0 principles, which included a near field communication platform to improve consumer service. Capaci et al. [32] presented an innovative performance monitoring system, specifically devoted to control loops, based on cloud technology by focusing on three different aspects: describing the entire cloud architecture and its implementation issues, illustrating basic techniques and features installed in the updated analytics tool, and presenting significant case studies. The system illustrated a successful example of a cloud-based platform for performance monitoring and assessment of process plants specifically oriented to proportional-integral-derivative control loops. Capaci et al. [33] designed and tested an automated system for modeling and controlling color quality of dyed lathers. The proposed software was fully integrated with the machineries of the finishing line and an automated tintometer system. A set of company data was used to validate the identified colorimetric models and the proposed color correction strategy. The paradigm shifts in manufacturing that Industry 4.0 brings forth with new advanced technologies and the rapid growth of sensing and controlling technologies enable further visualization and optimization that can contribute to achieving improved decision-making in manufacturing. A significant new capability is the ability to construct a Digital Twin that connects the physical and virtual space [34]. Z. Han et al. [35,36] explored the multi-queue limited buffers scheduling problems in a flexible flow shop with setup times in a bus manufacturer and designed several local scheduling rules to control the moving process of the work pieces.

In a previous published study, a mathematical model was developed for the yogurt filling system with different processing layout and process parameters [37]. The main objective of model was to maximize the speed of the conveyor belt within the allowable limit. This speed was linked with the feed rate of nozzles of the yogurt and flavor valves. Increasing the feed rates of the nozzles increased the speed of the conveyor belt and vice versa. Two different points over the conveyor belt were considered for filling the yogurt and flavors into cups of different volumes demanded by customers. The total length of the conveyor belt was divided into three equal segments. The three segments included the distance from the entry point to the yogurt filling point, the distance from the yogurt filling point to the flavor filling point, and the distance from the flavor filling point to the exit

point. In the mathematical model, the decision variables considered were the feed rates of yogurt and flavor valves linked with the speed of the conveyor belt.

In the present article, the setting of the yogurt filling machine was changed from multi filling points (Case-I) to a single filling point (Case-II) to check which setting results in more intended outputs. A mathematical model was developed for the new machine setting with an objective to maximize the throughput or minimize the processing time and the results were used as inputs in the single-dimension rules for the purpose to select a better setting in Case-I and Case-II. Additionally, the outcomes of the performance measures for Case-I and Case-II were compared using single-dimension rules (EDD, SPT, or FCFS) to find a rule which results in a better performance measure preferred by the management.

The article is divided into different sections as follows. In Section 1, an introduction including a literature survey of the research work is provided while the problem description is written in Section 2, where operational and technological constraints as well as the different assumptions for both Case-I and Case-II are presented. In Section 3, the methodology adopted for the process optimization problem, the possible combinations of yogurt mixing with different flavors, and the equations for different processing times are explained in detail. Section 4 illustrates the solution procedure of the problem and a customer order problem is solved while Section 5 provides details of the sequencing of customer order processing based on the single-dimension rules. The results are discussed in Section 6 while conclusion and future research directions appear in Section 7.

## 2. Problem Description

The research work presented in this article focused on the process optimization of a yogurt and flavor filling machine based on Industry 4.0 concept. The speed of the belt is linked with the feed rates of the nozzles of the valves. The speeds of the conveyor belts carrying either the empty, or only yogurt-filled, or yogurt mixed with any flavor-filled cups and feed rates of the yogurt and flavor nozzle valves were determined to minimize the filling process time. The cups were filled with required volumes of yogurt and different flavors according to the customer demands in the minimum possible time. In this study, the following assumptions and constraints were considered:

### 2.1. Operational Constraints (Management and Customers' Specifications)

1. Consideration of the minimum cup volume to optimize the sequence of processing the cups.
2. Satisfaction of customer specifications with minimum and maximum yogurt volumes with three different flavors.
3. Filling of the same volumes of yogurt mixed with required flavors in batches.
4. Filling of yogurt and flavors at a single point results in mixed yogurt and flavors, while filling yogurt and flavors at different points results in flavors above the yogurt.

### 2.2. Technological Constraints (Machinery Characteristics)

1. Using minimum and maximum capacity cups.
2. Limited number of yogurt and flavor nozzles.
3. Minimum and maximum feed rates of the valves of yogurt and different flavors.
4. Limited number of conveyor belts at the machines.
5. Minimum and maximum speed of the conveyor belt carrying the cups.
6. Limited number of flavors and yogurt types.
7. Limited number of holders for cups over the conveyor belt.

There may be some standard tolerances in use in the industry of yogurt and flavor filling, which must be considered while fulfilling an order. The customers are obliged to accept deviations of the quantity ordered in specific ranges and in cases of over-production, the marketing department negotiates its acceptance by the consumer. During the planning phase, under-production is never considered due to the losses inherent in production.

Two cases were considered. In Case-I, the yogurt and flavor filling points are at different locations while in Case-II, the yogurt and flavors are filled into the cup from a single location. In Case-I, an empty cup is entered into the system by placing it on the entry point. It moves over the belt to the yogurt filling nozzle and is filled with required volume of yogurt. After that, it covers some distance to reach the flavor filling point. Once the desired flavor of required volume is filled, the completely filled cup then moves towards the leaving point over the belt and leaves the filling system. In Case-II, an empty cup enters the system through the placement point, it then moves towards the yogurt and flavor filling point located at a single point over the belt in the machine, where the cup is filled with required volumes of yogurt and flavors simultaneously, and the cup filled with the desired volumes then moves towards the exit point to leave the system. A few assumptions were considered in both cases and are given below.

*2.3. Case-I Assumptions*

1. The cups move over the conveyor belt and the belt is divided into three equal segments.
2. The yogurt and flavors are filled at two different points.
3. Any of the three flavors can be filled in the cups at a single point after the yogurt is filled.
4. There are five distinct times in the processing of cups: the traveling time of cup from the entry point to the yogurt filling point, the yogurt filling time, the time in which the only yogurt-filled cup moves towards the flavor filling point, the flavor filling time, and the time in which a completely filled cup moves to the exit point.
5. There are two conveyor belts moving in parallel and carrying cups between the entry and exit points. When one belt stops for the filling process, the other one remains in motion and moves the cups.
6. A cup is filled with yogurt and only one type of flavor.

*2.4. Case-II Assumptions*

1. The conveyor belt is divided equally into two segments.
2. The filling process of yogurt and all three different flavors is performed at a single point.
3. There are three equal times: the traveling time of cups from placement to the filling point, the time in which a cup is filled simultaneously with the required volumes of yogurt and flavors, and the time in which the completely filled cup travels from the filling point to the exit point.
4. There are two conveyor belts which move in parallel and when one stops for the filling process, the other one carries the cups between the entry and exit points.
5. A cup can be filled with yogurt and multiple flavors.

There were a few other assumptions considered in both cases. The common assumptions include that the processing is uninterrupted, no cancellation and arrival of orders occur once the filling process has started, and the processing times are deterministic.

**3. Production Line Architecture**

Although a mathematical model for the process optimization of the Industry 4.0-based yogurt filling machine was developed in the current article, the details of the architecture of production line also needs to be discussed. The machine setting was changed from multiple filling points to a single filling point and a straight conveyor belt was considered. In the previous setting, the arrangement of the conveyor belt was either L- or U-shaped.

In the previous setting, the conveyor belt was semi-automatic and human intervention was often required for the production process completion. The semi-automatic conveyor belt is now shifted to complete the automatic mode, which increased the throughput and reduced the process time by using the latest technology IR 4.0 enablers.

For flavor identification in the previous machine setting, each cup had a color code. Additionally, for quantity grouping, optical sensors were used. In the new setting, the Near-Field Communication (NFC) technology was used for quantity categorization and color identification. NFC tags comprising of all details replaced the different colored bottles with uniform bottles from the production line.

The Fanuc LR Mate 200ic robotic arm was used for loading and unloading of NFC-tagged empty cups and yogurt-filled cups, respectively. The robotic arm was programmed in a way such that it could gently lift the empty cups and place them on the entry point of the machine. It was also used for the removal of the completely filled cups from the exit point of the machine and placing them outside the system.

In the new setting, all filling nozzles were unified into a single head which was designed in solidworks software. The drilling and milling operations were performed in the laboratory to make the unified head from a stainless steel metal block. The block was used to connect the filling nozzles with the yogurt and flavor containers through tubes.

A PN532 NFC/RFID controller breakout board manufactured by Adafruit was used for reading and writing data. The Future Technology Devices International (FTDI) chip was used for the interfacing of devices and to power the PN532 board. The information pasted on the empty cups transforms with the motion of the cup to the conveyer belt. The tag information pasted on the empty cups is decoded by the PN532 NFC tag reader, transmitted to the Raspberry Pi, and commands are given to the WAGO Programmable Field Controller (PFC). The Raspberry Pi was programmed in such a way that it could provide digital control signals for defining the customer order.

To improve the functionality of the machine, a few new devices and components were added to it. In order to maintain a steady flow rate of the yogurt and flavors, diaphragm pumps were installed between the unified head of nozzles and the yogurt and flavors tanks. Electrically controlled solenoid valves were used to release the yogurt and flavors for a certain time depending on the feed rates of the valves. To avoid any spillage of the yogurt and flavors, a pneumatic manipulator arm was used for the alignment of the filling nozzle position in front of the cups. Additionally, a push button was used to start and stop the conveyor belt when needed.

The details about the major parts used in the automatic yogurt filling machine are provided in Table 1 below.

**Table 1.** Major parts used in the automatic yogurt filling machine.

| Part | Reason | Quantity |
|---|---|---|
| Piston | One for the NFC station, one for the filling station, and two for the start and end stations | 4 |
| Photoelectric Proximity Sensor | One for the NFC station, one for the filling station, and two for the start and end stations | 4 |
| Proximity Sensor Normally Closed | For the notification of process finishing | 1 |
| Solenoid Valve | Three for the flavors and one for the yogurt | 4 |
| Diaphragm Pump | One for each solenoid valve | 4 |
| Solid State Relay | For switching of the load that comes from the WAGO PFC | 4 |
| WAGO PFC | Managing the system | 1 |
| NFC Module | Read tags | 1 |
| Raspberry Pi | To control the signal from NFC | 1 |
| 5.5-inch OLED Touch Screen | To display the function of the Raspberry Pi | 1 |
| FTDI Chipped Board | To supply the power to the NFC Module | 1 |
| Switched Mode Power Supply (787–1602) | Switch the power source of the diaphragm WAGO controller (only 0.5 A) | 1 |
| Switched Mode Power Supply (787–1717) | Switch the power source of the diaphragm pump (only 2.0 A) | 1 |
| Stack Light | To provide the visual status for the system | 1 |
| Control Panel | For protection of the electrical and electronics parts | 1 |
| Air Filter Regulator | To filter the air coming from the air laboratory source | 1 |
| Tank | Three for the flavors and one for the yogurt | 4 |

### 4. Mathematical Modeling

The development of the mathematical model considers the process optimization of the overall system. Two processes were performed once a cup had been placed at the entry point and until it left the system. These two are the filling process and the movement of cups over the conveyor belt between any points. The speed of the belt is directly linked with the feed rates of the yogurt and flavor valves. The speed of the conveyor belt could be maximized within allowable limits, which resulted in an increased throughput. For every valve, there was a maximum feed rate value beyond which the feed rate could not be exceeded. In Figure 1 below, the previous setting in which the filling processes were performed at two distinct locations and the conveyor belt was divided into three equal segments (Case-I) can be seen.

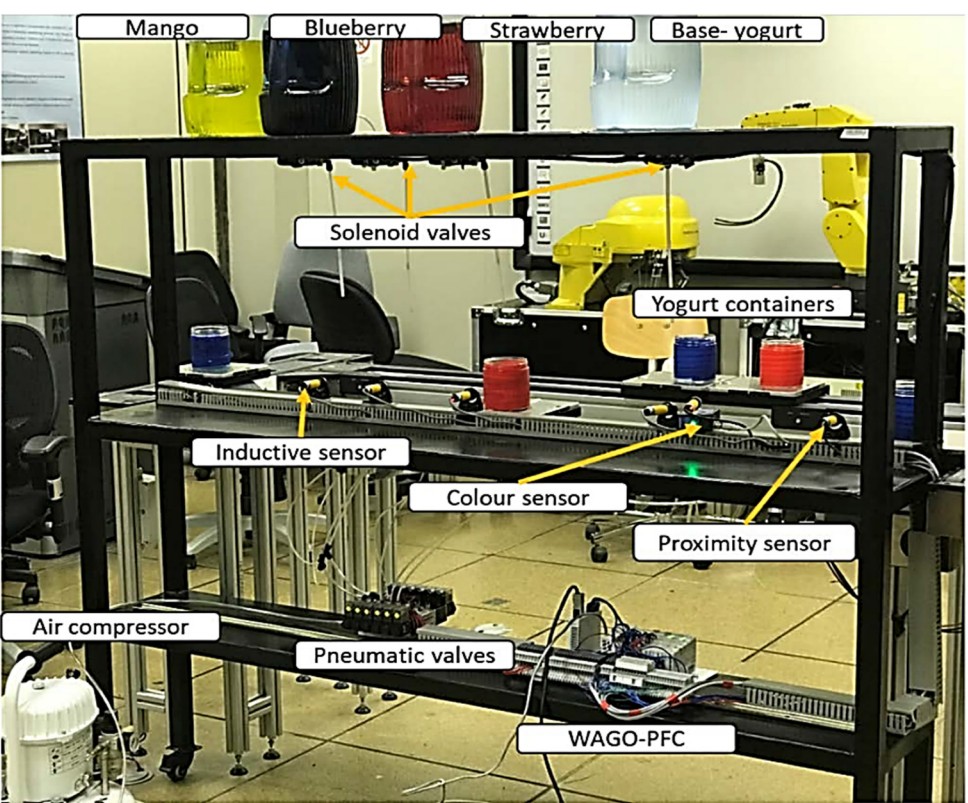

**Figure 1.** The yogurt filling machine with filling points at two distinct locations (Case-I). **Figure source:** B. Salah et al. [15].

A linear programming mathematical model was developed for the yogurt filling machine with the objective to maximize the speed of the conveyor belt and hence throughput, while also filling the cups with the required volumes of yogurt and different flavors to meet the customer demand in a given time. A few constraints were considered during the filling process. Some indices, parameters, and decision variables were considered in the mathematical modeling for process optimization and are given below.

*Indices*

| | | |
|---|---|---|
| $i$ | yogurt percentage | $i = 1, 2, \ldots, I$ |
| $y$ | yogurt type | $y = 1, 2, \ldots, Y$ |
| $j$ | flavor percentage | $j = 1, 2, \ldots, J$ |
| $f$ | flavor type | $f = 1, 2, \ldots, F$ |
| $k$ | total volume | $k = 1, 2, \ldots, K$ |

*Process Parameters*

$S_b$　conveyor belt speed
$L_t$　conveyor belt total length
$V_{iyjfk}$　yogurt volume in a total volume
$v_{iyjfk}$　flavor volume in a total volume
$D_{iyjfk}$　demand from customers
$W_{iyjfk}$　pickup time in minutes
$V_{max}$　upper limit of total volume of yogurt and flavors
$V_{min}$　lower limit of total volume of yogurt and flavors

*Decision variables*

$\beta_{iyjfk}$　yogurt valve feed rate
$\gamma_{iyjfk}$　flavor valve feed rate

Mathematically, the objective function for maximizing the speed of conveyor belt in terms of length between any two points on the conveyor belt, feed rate of the yogurt valve, and the required yogurt volume can be stated as follows:

$$\textbf{Maximize: } Z = l\sum_{i=1}^{I}\sum_{y=1}^{Y}\sum_{j=1}^{J}\sum_{f=1}^{F}\sum_{k=1}^{K}\left(\frac{\beta_{iyjfk}}{V_{iyjfk}}\right) \tag{1}$$

As the ratio of the feed rate of the yogurt solenoid valve to the yogurt volume and the ratio of the feed rate of the flavor solenoid valve to the flavor volume are equal, the objective function for the speed of the conveyor belt can also be written as follows:

$$\textbf{Maximize: } Z = l\sum_{i=1}^{I}\sum_{y=1}^{Y}\sum_{j=1}^{J}\sum_{f=1}^{F}\sum_{k=1}^{K}\left(\frac{\gamma_{iyjfk}}{v_{iyjfk}}\right) \tag{2}$$

To manage the movement of cups over the belt, five times were considered equal in Case-I. These times are the movement time of an empty cup from the placement to the yogurt filling point, the yogurt filling time, the movement time of the yogurt-filled cup from the yogurt to flavor filling points, the flavor filling time, and the movement time of the cup filled with yogurt and flavors from the flavor filling to exit points. Due to these equal times, the ratio of the yogurt feed rate to required volume of yogurt remained equal to the ratio of flavor feed rate to the required volume of flavor.

A few constraints and equations were considered in the model relevant to the speed of the conveyor belt, the feed rates of the valves, customer waiting time, and volumes of yogurt and flavors. These constraints and equations are given as follows:

$$\frac{\beta_{iyjfk}}{V_{iyjfk}}l \leq Maximum\ S_b \quad I = 1,2, \ldots, I \quad y = 1,2, \ldots, Y \quad j = 1,2, \ldots, J \quad f = 1,2, \ldots, F \quad k = 1,2, \ldots, K \tag{3}$$

$$\frac{\gamma_{iyjfk}}{v_{iyjfk}}l \leq Maximum\ S_b \quad i = 1,2, \ldots, I \quad y = 1,2, \ldots, Y \quad j = 1,2, \ldots, J \quad f = 1,2, \ldots, F \quad k = 1,2, \ldots, K \tag{4}$$

$$\beta_{iyjfk} \leq Maximum\ \beta_{iyjfk} \quad i = 1,2, \ldots, I \quad y = 1,2, \ldots, Y \quad j = 1,2, \ldots, J \quad f = 1,2, \ldots, F \quad k = 1,2, \ldots, K \tag{5}$$

$$\gamma_{iyjfk} \leq Maximum\ \gamma_{iyjfk} \quad i = 1,2, \ldots, I \quad y = 1,2, \ldots, Y \quad j = 1,2, \ldots, J \quad f = 1,2, \ldots, F \quad k = 1,2, \ldots, K \tag{6}$$

$$\frac{60W_{iyjfk}\beta_{iyjfk}}{V_{iyjfk}} - 4 \geq D_{iyjfk} \quad i = 1,2, \ldots, I \quad y = 1,2, \ldots, Y \quad j = 1,2, \ldots, J \quad f = 1,2, \ldots, F \quad k = 1,2, \ldots, K \tag{7}$$

$$\frac{V_{iyjfk}}{\beta_{iyjfk}} = \frac{v_{iyjfk}}{\gamma_{iyjfk}} \quad i = 1,2, \ldots, I \quad y = 1,2, \ldots, Y \quad j = 1,2, \ldots, J \quad f = 1,2, \ldots, F \quad k = 1,2, \ldots, K \tag{8}$$

The objective functions (1) and (2) of the model resulted in equal values and maximized the conveyor belt speed. The constraints (3) and (4) were for the conveyor belt speed while the inequalities (5) and (6) were used for the feed rates of yogurt and flavor valves, respectively. Constraint (7) is written to make sure that the processing time is less than or

equal to the customer waiting time, whereas Equation (8) satisfies that the yogurt filling time should be equal to the flavor filling time.

The possibility of processing time minimization was noted in the yogurt filling machine setting (Case-I). It was decided to change the setting and the filling operation of yogurt and flavors to be performed at a single location over the conveyor belt (Case-II). The modified system with yogurt and flavor filling points at a single location (Case-II) can be seen in Figure 2.

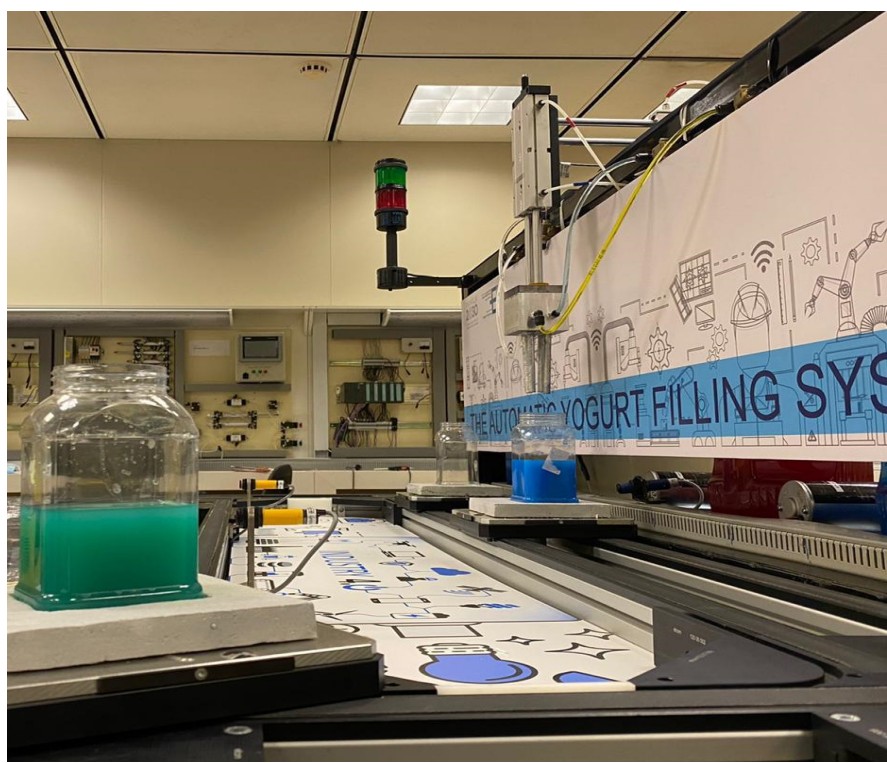

**Figure 2.** Modified setting of the yogurt filling system (Case-II).

All other constraints for Case-II remained the same as those for Case-I, except constraint (7), which was related to the customer waiting time for filling the required number of cups with the desired volumes of yogurt and flavors. Constraint (7) can be changed to constraint (9) for the new setting (Case-II) and can be written as follows:

$$\frac{60W_{iyjfk}\beta_{iyjfk}}{V_{iyjfk}} - 2 \geq D_{iyjfk} \quad i = 1, 2, \ldots, I \quad y = 1, 2, \ldots, Y \quad j = 1, 2, \ldots, J \quad f = 1, 2, \ldots, F \quad k = 1, 2, \ldots, K \quad (9)$$

As already discussed in Section 2, there is a constraint of maximum and minimum total volume of a cup and orders for greater and lower volumes than the available sizes of cups must not be accepted. To optimize the use of the available machine, the constraints (10) are written in one line for both minimum and maximum limits of the total volume of yogurt and flavors in a cup.

$$V_{min} \leq (V_{iyjfk} + v_{iyjfk}) \leq V_{max} \quad i = 1, 2, \ldots, I \quad y = 1, 2, \ldots, Y \quad j = 1, 2, \ldots, J \quad f = 1, 2, \ldots, F \quad k = 1, 2, \ldots, K \quad (10)$$

The $v_{iyjfk}$ is used for the volume of flavors, but it can be either zero or non-zero. The zero value of the $v_{iyjfk}$ means customer demand for pure yogurt and non-zero value of the $v_{iyjfk}$ means flavor(s) mixed with yogurt.

Depending on the customer demand, either a single flavor or a combination two or more than two flavors can be mixed with the yogurt. All customer orders for pure

yogurt or a combination of flavors mixed with yogurt can be found by the formula given in Equation (11).

$$\text{Number of orders for yogurt or combination of flavors mixed with yogurt} = 2^n \quad (11)$$

where $n$ is the number of available flavors.

All customer orders for yogurt and three available flavors are summarized in Table 2. In all customer orders, yogurt is demanded and hence in all combinations its value is 1. A value of 1 for a flavor in a combination means the presence of that flavor. The customer may demand pure yogurt, which is the combination in which yogurt takes the value of 1 and all three flavors are 0 while yogurt mixed with all three flavors is a combination in which the yogurt as well as all three flavors have the value of 1. The percentages of yogurt and flavors in a total volume also depend on the customer demand and can take any value (in percentages) provided that the minimum total volume is greater than or equal to the lower limit and less than or equal to the upper limit.

**Table 2.** All combinations of yogurt mixing with three flavors.

| Yogurt | Flavor 1 | Flavor 2 | Flavor 3 | $V_{min}$ (mL) | $V_{max}$ (mL) |
|--------|----------|----------|----------|----------------|----------------|
|   | 0 | 0 | 0 |   |   |
|   | 1 | 0 | 0 |   |   |
|   | 0 | 1 | 0 |   |   |
|   | 1 | 1 | 0 |   |   |
| 1 |   |   |   | 250 | 1000 |
|   | 0 | 0 | 1 |   |   |
|   | 1 | 0 | 1 |   |   |
|   | 0 | 1 | 1 |   |   |
|   | 1 | 1 | 1 |   |   |

While placing the very first cup on the entry point of the machine, there is no waiting time for the cup to enter into the system while the waiting time of the second cup for placement is equal to the time in which the first cup reaches from the entry point to the yogurt or flavor filling point. Similarly, the waiting time of the third cup to enter into the system in the sum of the filling time of the very first cup and the movement time of the second cup from the entry point to the filling point. A general relation for the waiting time of the $n$th empty to enter into the system is given in Equation (12) in terms of required volume and feed rate of the yogurt valve.

$$E_n = (n-1)\frac{V_{iyjfk}}{\beta_{iyjfk}} \quad i = 1,2,\ldots,I \quad y = 1,2,\ldots,Y \quad j = 1,2,\ldots,J \quad f = 1,2,\ldots,F \quad k = 1,2,\ldots,K \quad (12)$$

The two processes performed in the machine to fulfill customer orders are the movement of the cups over the conveyor belt and the filling of cups. In Case-I, the filling processes and movements of cups are performed in equal times. There are two filling processes and three movements of a cup in the machine; hence, the time taken by a cup to move from the entry to the exit point and complete the filling process is given in Equation (13).

$$P = 5\frac{V_{iyjfk}}{\beta_{iyjfk}} \quad i = 1,2,\ldots,I \quad y = 1,2,\ldots,Y \quad j = 1,2,\ldots,J \quad f = 1,2,\ldots,F \quad k = 1,2,\ldots,K \quad (13)$$

The very first cup, when moving from the entry to the exit point, takes time in its movement and filling processes. As there are three movements and two filling processes and all are performed in equal times, the first cup reaches the exit point in the sum of five equal times. The waiting time for the second cup until it reaches the exit point is the sum of the five equal times and the time in which the second cup reaches from the filling point to

the exit point. A general relation to wait for the $n$th completely filled cup at the exit point is given in Equation (14) in terms of the required volume and feed rate of the yogurt valve.

$$F_n = (n+4)\frac{V_{iyjfk}}{\beta_{iyjfk}} \quad i = 1, 2, \ldots, I \quad y = 1, 2, \ldots, Y \quad j = 1, 2, \ldots, J \quad f = 1, 2, \ldots, F \quad k = 1, 2, \ldots, K \tag{14}$$

It can be seen in the above equations that the ratio of the required yogurt volume and the yogurt feed rate was used. Instead of this, the ratio of the required flavor volume and flavor feed rate can also be used, as both the ratios are considered equal in the model, as can be seen in Equation (8).

Similarly, for Case-II, Equation (12) was used for the waiting of the $n$th empty cup to enter the machine for the movement and filling processes of cups, while Equation (15) shows the total time taken by a cup from the entry to the exit point in the machine. Equation (16) is used to find the waiting time for a cup at the exit point of the machine.

$$P = 3\frac{V_{iyjfk}}{\beta_{iyjfk}} \quad i = 1, 2, \ldots, I \quad y = 1, 2, \ldots, Y \quad j = 1, 2, \ldots, J \quad f = 1, 2, \ldots, F \quad k = 1, 2, \ldots, K \tag{15}$$

$$F_n = (n+2)\frac{V_{iyjfk}}{\beta_{iyjfk}} \quad i = 1, 2, \ldots, I \quad y = 1, 2, \ldots, Y \quad j = 1, 2, \ldots, J \quad f = 1, 2, \ldots, F \quad k = 1, 2, \ldots, K \tag{16}$$

It can be noted that Equations (12) was used in both Case-I and Case-II, as the waiting time of the cups to enter the system can be the same. The difference can be seen in the processing time and waiting time at the exit point of the machine during the filling process of the cups.

The benefits of the proposed model are as follows.

- Less computational burden while solving through software packages
- It is used for the flavor valve's size selection by finding the maximum feed rate of the flavor valve
- It is used for finding the optimal solution and satisfying all constraints
- It is used for finding the maximum values of the speed of the conveyor belt and feed rates of the yogurt and flavor nozzles
- It is used to find the processing rate of cups demanded by customers

### 5. Solution Procedure

The solution procedure adopted for the problem is divided into several stages. First, the upper limits of the conveyor belt speed and the feed rates of the yogurt and flavor valves were evaluated. The total length of the conveyor belt and the length between any two points (entry, yogurt and flavor filling, and exit point) were calculated. As orders from customers are received in the form of required volumes of yogurt and flavors in the total volume of a cup and the total number of cups, all values were put in the model and the objective function and constraints were set to determine the feasibility of the orders. The customer order must satisfy the upper- and lower-volume limits. Once the model was solved through the Simplex algorithm and the optimal solution is found, the values of all measures (feed rates of yogurt of flavor valves, speed on the conveyor belt, etc.) were calculated and results were documented. The movement and filling of cups with the required volumes of yogurt and flavors was then started to meet the customer demand.

For illustration purposes, a real life problem previously solved using the Case-I model was solved again using the Case-II model. A problem comprising six customer orders was solved for both cases. The set of orders is described in Table 3. The customer demand was for yogurt of a single type and three different types of flavors. In the mathematical model, the yogurt is represented by 1 and the flavors, i.e., strawberry, blueberry, and mango, are represented by 1, 2, and 3, respectively. It is to be noted that orders 1 and 5, orders 2 and 3, and orders 4 and 6 are customer orders for yogurt mixed with strawberry, blueberry, and mango flavors, respectively. The set of orders must satisfy the upper and lower limits of 1000 mL and 250 mL, respectively, on the total volume of yogurt and flavors.

**Table 3.** Orders from customers.

| Order No. | Volume (mL) | Yogurt (%) | Flavor 1 (%) | Flavor 2 (%) | Flavor 3 (%) | $D_{iyjfk}$ (Units) | $W_{iyjfk}$ (Minutes) |
|---|---|---|---|---|---|---|---|
| 1 | 300 | 93 | 7 | 0 | 0 | 100 | 10 |
| 2 | 300 | 90 | 0 | 10 | 0 | 80 | 9 |
| 3 | 600 | 90 | 0 | 10 | 0 | 25 | 7 |
| 4 | 500 | 95 | 0 | 0 | 5 | 35 | 20 |
| 5 | 900 | 95 | 5 | 0 | 0 | 20 | 15 |
| 6 | 900 | 93 | 0 | 0 | 7 | 30 | 25 |

To solve the problem using Case-I and Case-II, the maximum speed, feed rate of the yogurt valve, and feed rate of the flavor valve considered were 10 cm/s, 50 mL/s, and 25 mL/s, respectively. In Case-I, the total length of the conveyor belt (90 cm) was divided into three equal parts and each part was equal to 30 cm. In Case-II, the total length of the conveyor belt (90 cm) was divided into two equal parts and each part was equal to 45 cm.

The objective functions and constraints were written for both cases and the models were solved simultaneously to determine the optimal values of all decision variables satisfying all constraints and maximizing the throughput. The solutions of the models which resulted in optimal values of the decision variables are presented in Table 4.

**Table 4.** Optimal solutions for Case-I and Case-II.

| Case No. | Order No. | $\beta_{iyjfk}$ (mL/s) | $\gamma_{iyjfk}$ (mL/s) | $S_b$ (cm/s) | $E_n$ (s) | $P$ (s) | $F_n$ (s) |
|---|---|---|---|---|---|---|---|
| I | 1 | 50 | 3.76 | 5.38 | 552.4 | 27.9 | 580.3 |
| | 2 | 50 | 5.56 | 5.56 | 426.6 | 27.0 | 453.6 |
| | 3 | 50 | 5.56 | 2.78 | 259.2 | 54.0 | 313.2 |
| | 4 | 50 | 2.63 | 3.16 | 323.0 | 47.5 | 370.5 |
| | 5 | 50 | 2.63 | 1.75 | 324.9 | 85.5 | 410.4 |
| | 6 | 50 | 3.76 | 1.79 | 485.5 | 83.7 | 569.2 |
| II | 1 | 50 | 3.76 | 5.38 | 552.4 | 16.74 | 569.16 |
| | 2 | 50 | 5.56 | 5.56 | 426.6 | 16.2 | 442.8 |
| | 3 | 50 | 5.56 | 2.78 | 259.2 | 32.4 | 291.6 |
| | 4 | 50 | 2.63 | 3.16 | 323.0 | 28.5 | 351.5 |
| | 5 | 50 | 2.63 | 1.75 | 324.9 | 51.3 | 376.2 |
| | 6 | 50 | 3.76 | 1.79 | 485.5 | 50.22 | 535.68 |

The output of the model showed that the feed rates of both yogurt and flavor valves were equal to or less than the maximum allowable values of the feed rates of the solenoid valves. Normally, the yogurt volume in a cup is demanded more than the flavor volume, and this higher feed rate of the yogurt valve is needed more than the flavor valve's feed rate. The feed rates vary according to the percentage of yogurt or flavors in a cup. It is to be noted that both the yogurt and flavors were filled into the cups in equal times.

The results also showed that the speed of the conveyor belt was less than the maximum allowable limit while fulfilling all the customer orders. The speed is linked directly with the feed rate of the solenoid valves. The speed of the belt could not be increased further once the maximum allowable value of the feed rate of the solenoid valves was reached.

All constraints of the model were satisfied. The ratio of the required volume to feed rate of yogurt is equal to the ratio of the required volume to feed rate of flavors in filling all

customer orders. Additionally, it can be noted that the customer waiting in all orders was more than the processing time of an order.

## 6. Sequencing the Processing of Orders

Sequencing is the order in which cups of required volumes are processed to meet customer demand. The processing can be performed easily in case of simple customer orders, but for large orders, the situation becomes complicated. The processing order is considered crucial, as it affects the machine idle time, waiting time of cups in queues, and customer waiting time. The single-dimension rules comprise one of the categories in priority sequencing which determines priority based on a single aspect such as arrival time, due date, or processing time.

The results of the problem solved for Case-I and Case-II in Section 4 were considered for single-dimension priority rules, as can be seen in Table 5. The same set of customer orders for both cases was considered with the time since an order is received. The processing time (in minutes) for each order was calculated and it can be noted that the processing time on each order was lower than the pickup time for each order.

**Table 5.** The processing and pickup times of the set of orders for Case-I and Case-II.

| Case No. | Order No. | Minutes Since Order Arrival | Processing Time (Minutes) | Pickup Time (Minutes) |
|---|---|---|---|---|
| I | 1 | 0 | 9.67 | 10 |
| | 2 | 1 | 7.56 | 9 |
| | 3 | 1 | 5.22 | 7 |
| | 4 | 0 | 6.18 | 20 |
| | 5 | 3 | 6.84 | 15 |
| | 6 | 2 | 9.49 | 25 |
| II | 1 | 0 | 9.49 | 10 |
| | 2 | 1 | 7.38 | 9 |
| | 3 | 1 | 4.86 | 7 |
| | 4 | 0 | 5.86 | 20 |
| | 5 | 3 | 6.27 | 15 |
| | 6 | 2 | 8.93 | 25 |

As can be seen in Tables 6–8, the sequence of the set orders was set according to the single dimension rules. The time since an order was received was noted and the first order to be processed started at time zero. The finish time is the sum of the starting time and processing time while flow time is the sum of finish time and time since an order arrived. The scheduled pickup time was provided by customers at the time when an order had been placed while actual pickup time is the maximum time in finish and scheduled pickup times. The minutes early or minutes past due is the difference between the scheduled pickup time and the finish time. It is to be noted that only positive values of the difference of minutes early and minutes past due were considered.

The order in which the set of customer orders can be processed in the EDD rule in both cases (Case-I and Case-II) can be seen in Table 6. According to the EDD rule for Case-I, order 3 was the one with the shortest pick up time (7 min). Hence, it was processed before all other orders. The processing of order 3 started at time zero and finished at 5.22 min. The flow time (6.22 min) is the summation of the finish time (5.22 min) and the time since the order had arrived (1 min). As the scheduled pickup time was 7 min and the finish time was 5.22 min, the order was prepared 1.78 min earlier than the scheduled pickup time. The last order to be processed in the EDD rule was order 6, with the longest pickup time (25 min). The processing of order 6 started at time 35.47 min with a processing time of 9.49 min

and finished at 44.96 min. The flow time (46.96 min) is the summation of the finish time (44.96 min) and the time since the order had arrived (2 min). As the scheduled pickup time was 25 min and the finish time was 44.96 min, the order was prepared 19.96 min later than the scheduled pickup time. Similarly, for Case-II, order 3 was prepared 2.14 min earlier and order 6 was prepared 17.79 min later than the scheduled pickup times.

**Table 6.** Sequence of orders based on the EDD rule.

| Case No. | Order No. | Minutes Since Order Arrival | Starting Time | Processing Time | Finish Time | Flow Time | Scheduled Pickup Time | Actual Pickup Time | Minutes Early | Minutes Past Due |
|---|---|---|---|---|---|---|---|---|---|---|
| | 3 | 1 | 0.00 | 5.22 | 5.22 | 6.22 | 7 | 7.00 | 1.78 | - |
| | 2 | 1 | 5.22 | 7.56 | 12.78 | 13.78 | 9 | 12.78 | - | 3.78 |
| I | 1 | 0 | 12.78 | 9.67 | 22.45 | 22.45 | 10 | 22.45 | - | 12.45 |
| | 5 | 3 | 22.45 | 6.84 | 29.29 | 32.29 | 15 | 29.29 | - | 14.29 |
| | 4 | 0 | 29.29 | 6.18 | 35.47 | 35.47 | 20 | 35.47 | - | 15.47 |
| | 6 | 2 | 35.47 | 9.49 | 44.96 | 46.96 | 25 | 44.96 | - | 19.96 |
| | 3 | 1 | 0.00 | 4.86 | 4.86 | 5.86 | 7 | 7.00 | 2.14 | - |
| | 2 | 1 | 4.86 | 7.38 | 12.24 | 13.24 | 9 | 12.24 | - | 3.24 |
| II | 1 | 0 | 12.24 | 9.49 | 21.73 | 21.73 | 10 | 21.73 | - | 11.73 |
| | 5 | 3 | 21.73 | 6.27 | 28.00 | 31.00 | 15 | 28.00 | - | 13.00 |
| | 4 | 0 | 28.00 | 5.86 | 33.86 | 33.86 | 20 | 33.86 | - | 13.86 |
| | 6 | 2 | 33.86 | 8.93 | 42.79 | 44.79 | 25 | 42.79 | - | 17.79 |

Where required, the unit of time is "minutes".

**Table 7.** Sequence of the orders based on the SPT rule.

| Case No. | Order No. | Minutes Since Order Arrival | Starting Time | Processing Time | Finish Time | Flow Time | Scheduled Pickup Time | Actual Pickup Time | Minutes Early | Minutes Past Due |
|---|---|---|---|---|---|---|---|---|---|---|
| | 3 | 1 | 0.00 | 5.22 | 5.22 | 6.22 | 7 | 7.00 | 1.78 | - |
| | 4 | 0 | 5.22 | 6.18 | 11.40 | 11.40 | 20 | 20.00 | 8.60 | - |
| I | 5 | 3 | 11.40 | 6.84 | 18.24 | 21.24 | 15 | 18.24 | - | 3.24 |
| | 2 | 1 | 18.24 | 7.56 | 25.80 | 26.80 | 9 | 25.80 | - | 16.80 |
| | 6 | 2 | 25.80 | 9.49 | 35.29 | 37.29 | 25 | 35.29 | - | 10.29 |
| | 1 | 1 | 35.29 | 9.67 | 44.96 | 45.96 | 10 | 44.96 | - | 34.96 |
| | 3 | 1 | 0.00 | 4.86 | 4.86 | 5.86 | 7 | 7.00 | 2.14 | - |
| | 4 | 0 | 4.86 | 5.86 | 10.72 | 10.72 | 20 | 20.00 | 9.28 | - |
| II | 5 | 3 | 10.72 | 6.27 | 16.99 | 19.99 | 15 | 16.99 | - | 1.99 |
| | 2 | 1 | 16.99 | 7.38 | 24.37 | 25.37 | 9 | 24.37 | - | 15.37 |
| | 6 | 2 | 24.37 | 8.93 | 33.30 | 35.30 | 25 | 33.30 | - | 8.30 |
| | 1 | 1 | 33.30 | 9.49 | 42.79 | 43.79 | 10 | 42.79 | - | 32.79 |

Where required, the unit of time is "minutes".

In the SPT rule, the order in which the set of customer orders can be processed in both cases (Case-I and Case-II) can be seen in Table 7. According to the SPT rule for Case-I, order 3 was the order with the shortest processing time (5.22 min). Hence, it was processed before all other orders. The processing of order 3 started at time zero and finished at 5.22 min. The flow time (6.22 min) is the summation of finish time (5.22 min) and time since the order had arrived (1 min). As the scheduled pickup time was 7 min and the finish time was 5.22 min, order 3 was prepared 1.78 min earlier than the scheduled pickup time. The last order to be processed in the SPT rule was order 1, with the longest processing time (9.67 min). The

processing of order 1 started at 35.29 min with a processing time of 9.67 min and finished at 44.96 min. The flow time (45.96 min) is the summation of the finish time (44.96 min) and the time since the order had arrived (1 min). As the scheduled pickup time was 10 min and the finish time was 44.96 min, the order was prepared 34.96 min later than the scheduled pickup time. Similarly, for Case-II, order 3 was prepared 2.14 min earlier and order 1 was prepared 32.79 min later than the scheduled pickup times.

**Table 8.** Sequence of orders based on the FCFS rule.

| Case No. | Order No. | Minutes Since Order Arrival | Starting Time | Processing Time | Finish Time | Flow Time | Scheduled Pickup Time | Actual Pickup Time | Minutes Early | Minutes Past Due |
|---|---|---|---|---|---|---|---|---|---|---|
|  | 5 | 3 | 0.00 | 6.84 | 6.84 | 9.84 | 15 | 15.00 | 8.16 | - |
|  | 6 | 2 | 6.84 | 9.49 | 16.33 | 18.33 | 25 | 25.00 | 8.67 | - |
| I | 3 | 1 | 16.33 | 5.22 | 21.55 | 22.55 | 7 | 21.55 | - | 14.55 |
|  | 2 | 1 | 21.55 | 7.56 | 29.11 | 30.11 | 9 | 29.11 | - | 20.11 |
|  | 4 | 0 | 29.11 | 6.18 | 35.29 | 35.29 | 20 | 35.29 | - | 15.29 |
|  | 1 | 0 | 35.29 | 9.67 | 44.96 | 44.96 | 10 | 44.96 | - | 34.96 |
|  | 5 | 3 | 0.00 | 6.27 | 6.27 | 9.27 | 15 | 15.00 | 8.73 | - |
|  | 6 | 2 | 6.27 | 8.93 | 15.20 | 17.20 | 25 | 25.00 | 9.80 | - |
| II | 3 | 1 | 15.20 | 4.86 | 20.06 | 21.06 | 7 | 20.06 | - | 13.06 |
|  | 2 | 1 | 20.06 | 7.38 | 27.44 | 28.44 | 9 | 27.44 | - | 18.44 |
|  | 4 | 0 | 27.44 | 5.86 | 33.30 | 33.30 | 20 | 33.30 | - | 13.30 |
|  | 1 | 0 | 33.30 | 9.49 | 42.79 | 42.79 | 10 | 42.79 | - | 32.79 |

Where required, the unit of time is "minutes".

Likewise, the order in which the set of customer orders can be processed in the FCFS rule in both cases (Case-I and Case-II) can be seen in Table 8. According to the FCFS rule for Case-I, order 5 was the one which arrived earlier than all the other orders (3 min since order arrival). Hence, it was processed before all other orders. The processing of order 5 started at time zero and finishes at 6.84 min. The flow time (9.84 min) is the summation of the finish time (6.84 min) and the time since the order arrived (3 min). As the scheduled pickup time was 15 min and the finish time was 6.84 min, the order was prepared 8.16 min earlier than the scheduled pickup time. The last order to be processed in the FCFS rule was order 1. The processing of order 1 started at 35.29 min with a processing time of 9.67 min and finished at 44.96 min. The flow time (44.96 min) was equal to finish time (44.96 min) as the time since the order had arrived was zero. As the scheduled pickup time was 10 min and the finish time was 44.96 min, the order was prepared 34.96 min later than the scheduled pickup time. Similarly, for Case-II, order 5 was prepared 8.73 min earlier and order 1 was prepared 32.79 min later than the scheduled pickup times.

In the three abovementioned single-dimension rules, the setup time was assumed to be independent of the processing sequence. Normally, this is not used and orders with similar setups are sequenced back-to-back in order to reduce the setup time. Some other assumptions considered during the problem solution using Case-I and Case-II were the deterministic filling time, uninterrupted processing, and no cancellation and arrival of new orders once the machine started to fill the cups with yogurt and flavors.

## 7. Results and Discussion

All information and data used in the solution of the problem using Case-I and Case-II were provided by the laboratory of the Computer Integrated Manufacturing (CIM), Department of Industrial Engineering, King Saud University, Saudi Arabia.

In Case-I and Case-II, the upper limit of the feed rate of the yogurt was considered as 50 mL/s and, hence, in both cases, the models resulted in the maximum allowable value (50 mL/s). The conveyor belt speed was controlled by the yogurt feed rate, which

constitutes a directly proportional relationship. In all orders, the flavor volume was lower than the yogurt volume in the total volume of an order and, hence, a lower feed rate of the flavor valve is needed to fill the cup with the required volume of flavor. In both cases, the speed of the conveyor belt and waiting time for a cup to enter the system remained the same. While comparing the two cases, it was found that the processing time of a cup and the total processing time of a set of customer orders were different.

In Figure 3, it can be seen that for each order, the total processing time in Case-I was greater than Case-II. This is because in Case-I, the yogurt and all flavors were filled at two distinct locations, while in Case-II, the filling of yogurt and flavors was at a single location. In both cases, the total processing time initially decreased from order 1 to order 3 and then increased from order 3 to order 6. In the case of order 1 and 2, the total volumes were equal, but the processing time in the case of order 1 was higher than that of order 2. This is due to the difference in the number of units demanded by the customer. In order 1, the number of units demanded (100 units) was greater than in order 2 (80 units). Similarly, in the case of order 5 and 6, the total processing time of order 6 was higher than that of order 5 for the same total volume (900 mL). This is because in order 6, the number of units demanded (30 units) was greater than the number of units demanded (20 units) in order 5. Hence, the total processing time of an order depends on the total volume and the number of units demanded. The greater the total volume and the number of units demanded, the greater the total processing time of an order. In the cases of order 1 and 6, the total processing times were higher than those of other orders. In order 1, the total number of cups needed (100 units) was higher than in the other orders, although the total volume was small (300 mL); in order 6, the total volume of a cup (900 mL) was higher than in the other orders, although the total number of cups (30 units) was smaller.

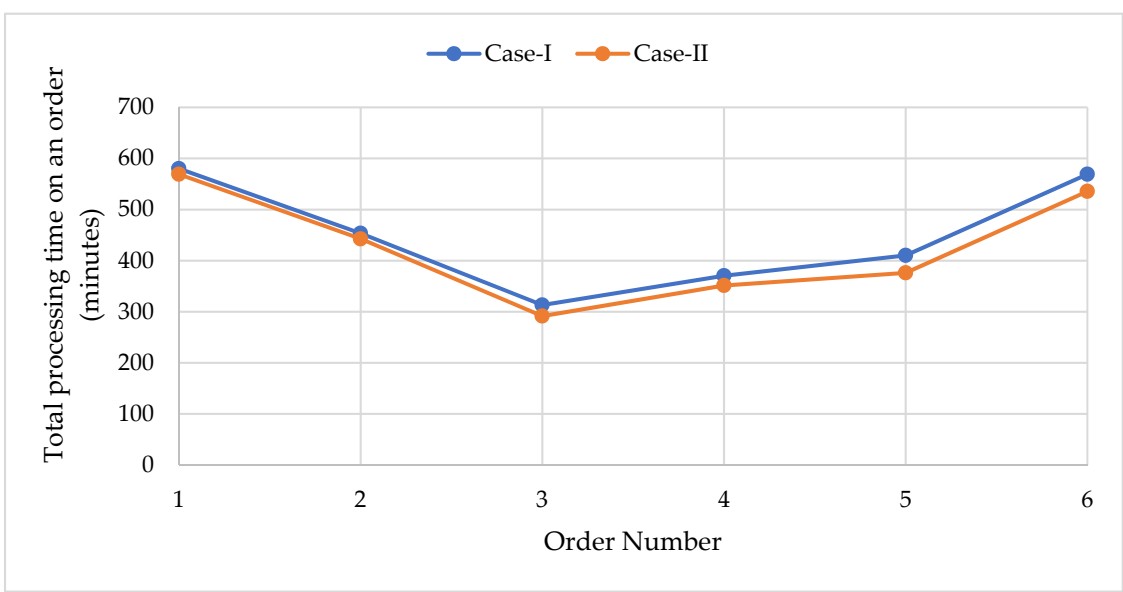

**Figure 3.** The total processing time on an order in Case-I and Case-II.

As shown in Table 3 in Section 5, $F_n$ is the total processing time of a set of orders. In Case-I, the processing times for orders 1, 2, 3, 4, 5, and 6 were 580.3 s, 453.6 s, 313.2 s, 370.5 s, 410.4 s, and 569.2 s, respectively. In Case-II, the processing times for orders 1, 2, 3, 4, 5, and 6 were 569.16 s, 442.8 s, 291.6 s, 351.5 s, 376.2 s, and 535.68 s, respectively.

By converting the times from seconds to minutes, the processing times in Case-I for orders 1, 2, 3, 4, 5, and 6 were 9.67 min, 7.56 min, 5.22 min, 6.18 min, 6.84 min, and 9.49 min, respectively. In Case-II, the processing times for orders 1, 2, 3, 4, 5, and 6 were 9.49 min, 7.38 min, 4.86 min, 5.86 min, 6.27 min, and 8.93 min, respectively.

$A$ = Sum of the processing times in Case-II = 42.78 min
$B$ = Sum of the processing times in Case-I = 44.95 min

$$\frac{B}{A} = 1.05 \text{ or } B = 1.05 \, A \tag{17}$$

Based on Table 9 and Equation (17), it can be stated that processing time of the machine in Case-II was 1.05 times faster than that of Case-I.

**Table 9.** Comparison of the processing times of Case-I and Case-II.

| Order No. | Processing Time in Case-II | Multiplicative Factor | Processing Time in Case-I |
|:---:|:---:|:---:|:---:|
| 1 | 9.49 | 1.02 | 9.67 |
| 2 | 7.38 | 1.02 | 7.56 |
| 3 | 4.86 | 1.07 | 5.22 |
| 4 | 5.86 | 1.05 | 6.18 |
| 5 | 6.27 | 1.09 | 6.84 |
| 6 | 8.93 | 1.06 | 9.49 |
| **Average of Multiplicative Factor** | | 1.05 | |

Where required, the unit of time is "minutes".

The set of customer orders was processed through Case-I and Case-II. It can be seen in Figures 4–6 that during the processing of each order, Case-II resulted in a lesser total processing time than in Case-I. Applying any single-dimension rules, the total time taken to process the set of orders in Case-I was 44.96 min, while in Case-II, it took 42.79 min. Hence, the results showed that Case-II was 1.05071 times faster than Case-I in processing the set of customer orders.

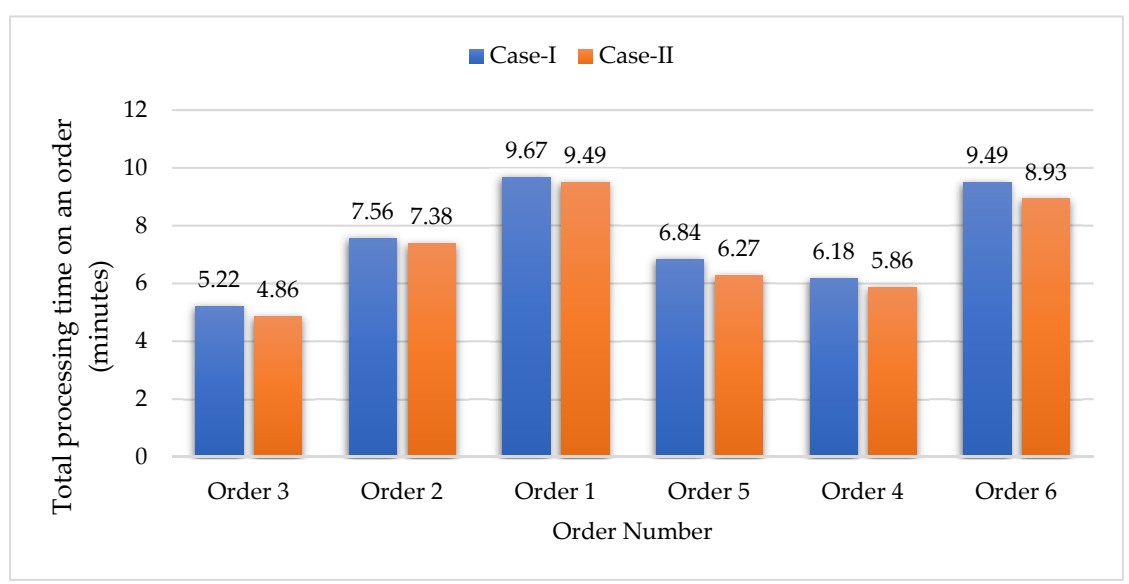

**Figure 4.** Comparison of the processing times for Case-I and Case-II using the EDD rule.

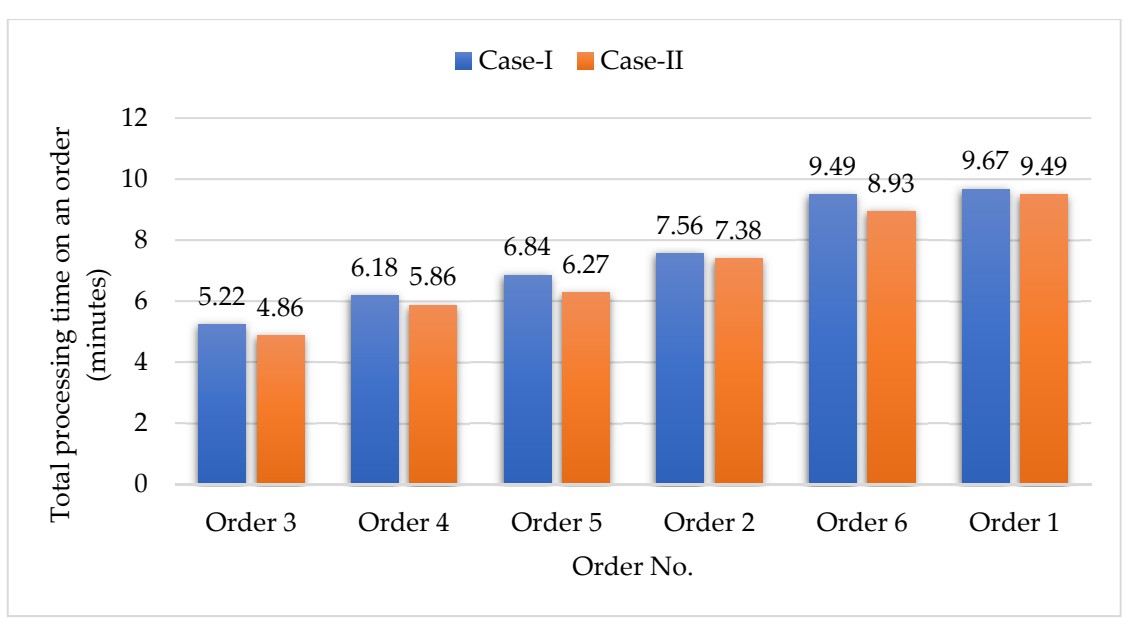

**Figure 5.** Comparison of the processing times for Case-I and Case-II using the SPT rule.

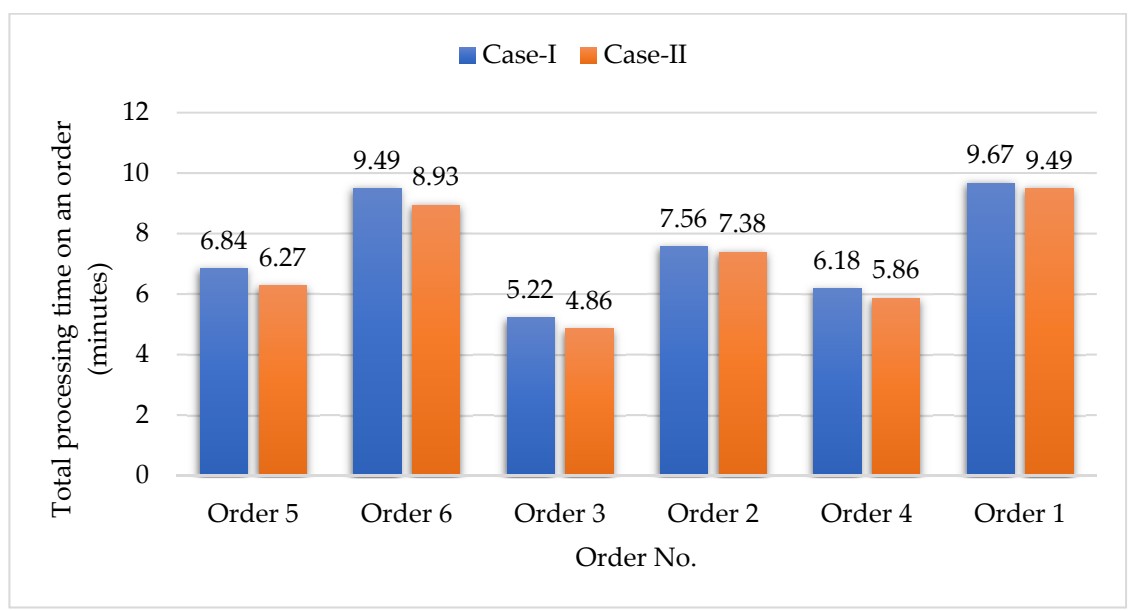

**Figure 6.** Comparison of the processing times for Case-I and Case-II using the FCFS rule.

The performance measures considered for the abovementioned priority sequencing rules for Case-I and Case-II were average flow time, average minutes early, and average minutes past due. The management tries to set the performance measures to the desired values and hence the minimum values of average flow time and average minutes past due, while the maximum value of average minutes early is preferred in the single-dimension rules when filling the cups with the required volumes of yogurt and flavors in Case-I and Case-II.

All the single-dimension rules resulted in slightly smaller values in Case-II than Case-I for average flow time and average minutes past due, while they resulted in slightly higher values in Case-II than in Case-I for average minutes early, as can be seen in Figures 7–9. In all single-dimension rules, the SPT rule resulted in minimum values of average flow time and average minutes past due performance measures, while the FCFS rule resulted in the highest values of average minutes early. It is to be noted that in all single-dimension

rules, Case-II resulted in better values than Case-I. Hence, the filling process was completed using Case-II.

Keeping promises to customers, the SPT rule provided better results than the EDD and FCFS rules with respect to average minutes past due and average flow time while the FCFS rule produced better results than the EDD and SPT rules with respect to average minutes early. As the management prefers improving the average minutes past due over the other performance measures, the SPT sequencing rule was used.

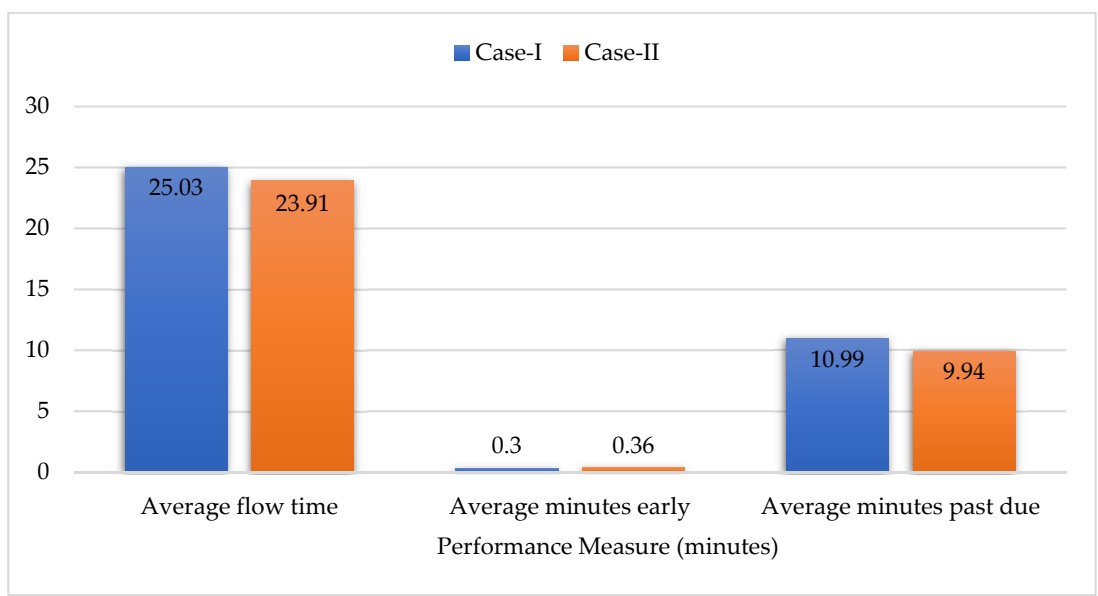

**Figure 7.** Comparison of the performance measures for Case-I and Case-II using the EDD rule.

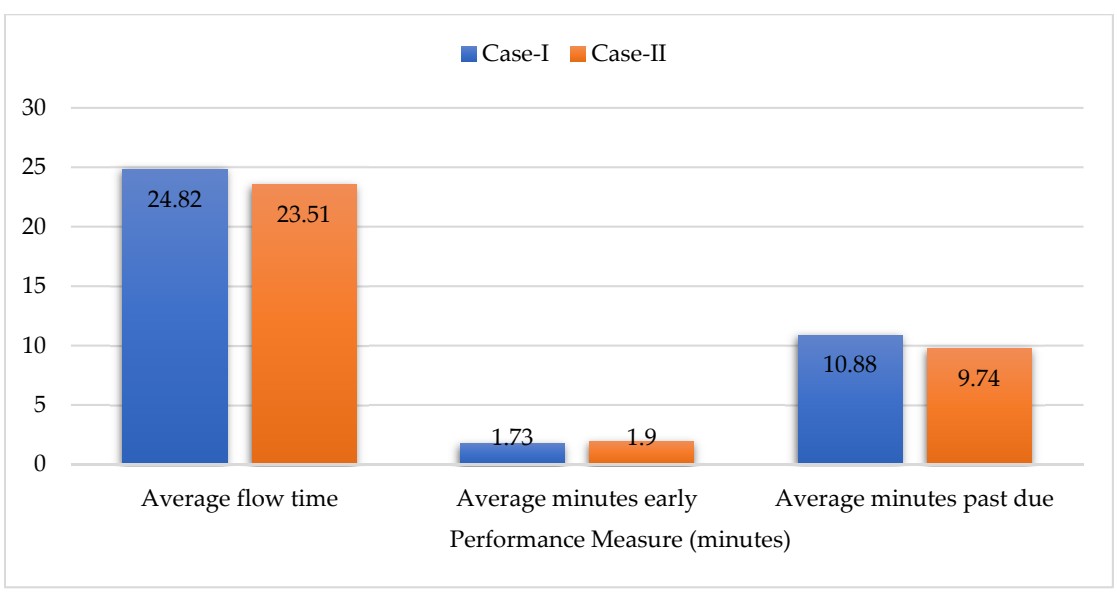

**Figure 8.** Comparison of the performance measures for Case-I and Case-II using the SPT rule.

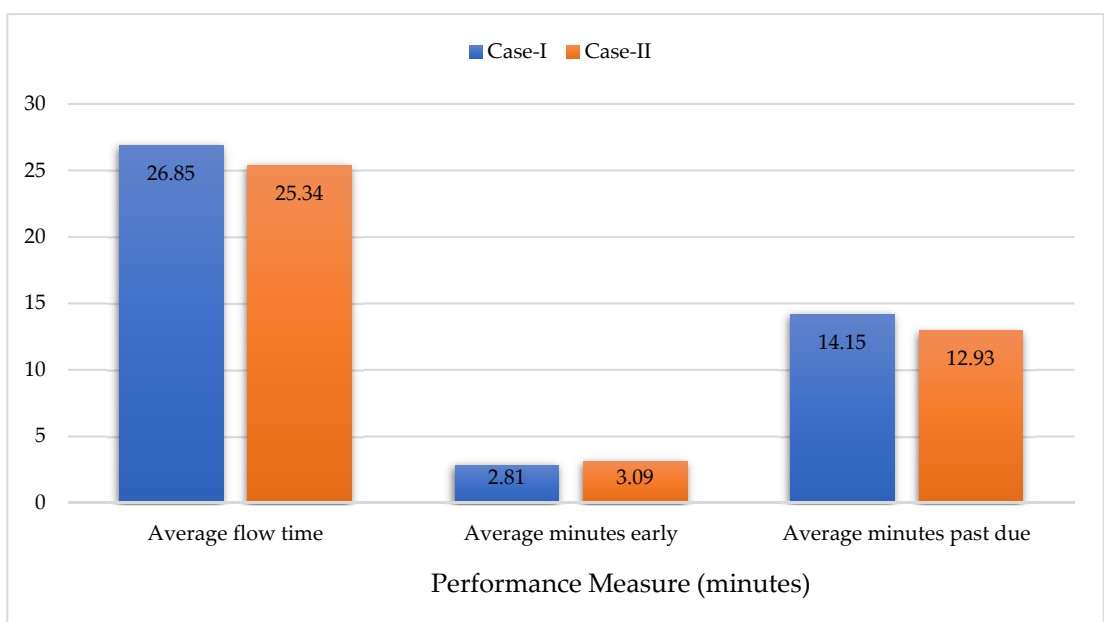

**Figure 9.** Comparison of the performance measures for Case-I and Case-II the using FCFS rule.

## 8. Conclusions

In this article, mathematical models for two settings (Case-I and Case-II) of an Industry 4.0-based yogurt filling system were developed and compared. The objectives of the models were the conveyor belt speed maximization or minimizing the processing time on an order or maximizing the throughput. In Case-I, the filling processes were performed at two different locations, while in Case-II, the filling processing was performed at a single location. The decision variables were the feed rates of the yogurt and flavor valves, and these variables controlled the speed of the conveyor belt.

The results showed that the processing time of an order in Case-II was lower than in Case-I due to the change in machine settings. Normally, the minimum customer waiting time is needed, and hence the machine setting in Case-II is preferred over the machine setting in Case-I. The results were also checked and it was found that Case-II resulted in the minimum total processing time using any single-dimension rule. Additionally, the outcomes of the performance measures for Case-I and Case-II were compared using single-dimension rules and it was found that SPT rule is used for the sequencing of orders as the management prefers improving the average minutes past due performance measure.

In the future, three cases can be compared to find which one provides the desired results. In Case-I, the yogurt- and flavor-filling operations are performed at two distinct locations. In Case-II, both filling operations are performed in a single location, while in Case-III, there will be a dedicated set of conveyor belts for the filling of each flavor and a base yogurt.

**Author Contributions:** Conceptualization, R.K. and B.S.; methodology, R.K. and B.S.; validation, J.C., Y.C. and Y.L.; formal analysis, W.S.; investigation, B.S.; resources, B.S.; writing—original draft preparation, R.K.; writing—review and editing, J.C., Y.C. and Y.L.; supervision, W.S.; project administration, R.K. All authors have read and agreed to the published version of the manuscript.

**Funding:** This study received funding from King Saud University, Saudi Arabia, through researchers supporting project number (RSP-2021/145). Additionally, the APCs were funded by King Saud University, Saudi Arabia, through researchers supporting project number (RSP-2021/145). This study also received funding from Teaching Research and Reform Fund from SDJU Grant No. 20216 and Shanghai Multidirectional Forging Engineering Technology Research Center Grant No. 20DZ2253200.

**Institutional Review Board Statement:** Not applicable.

**Informed Consent Statement:** Not applicable.

**Data Availability Statement:** The data presented in this study are available on request from the corresponding authors.

**Acknowledgments:** The authors extend their appreciation to King Saud University, Saudi Arabia for funding this work through researchers supporting project number (RSP-2021/145).

**Conflicts of Interest:** The authors declare no conflict of interest.

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
