# Peer review of "The Effect of Changes in Settings from Multiple Filling Points to a Single Filling Point of an Industry 4.0-Based Yogurt Filling Machine"

_processes, doi:10.3390/pr10081642_

Round 1
Reviewer 1 Report
Here are my comments. I think overall it is a great job.
1.Only one question. In this article, the author claims Case I and Case II are different in processing times using EDD rule, or STD rule. I wonder if the author runs any statistical testing to claim it is different or just random noise. For example, on page 19 "the total time taken to process the set of order in Case-I is 44.96 minutes while in Case-II, it takes 42.79 minutes. Hence, the results show that Case-II is 1.05071 times faster than Case-I in processing the set of customer orders." From a statistician's perspective, it could be due to random noise.
2.Please understand the language will devalue your work and readers will lose their interest when reading the article.
Here are sentences needed to either clarify, rephase or spell check.
-
Page 2: Do you mean human-machine interaction instead of man-machine interaction?
-
Page 12: “There is” not “the is” under the Table 1 first sentence.
-
Page 15, line 4. “Needed than then flavor feed”. Need to clarify what does it mean?
-
Page 2 “The bottles when reached the yogurt and flavor filling point were filled using the solenoid valves.” Maybe “When reached the yogurt and flavor filling point, the bottles were filled using XXX”
-
Page 2 “The whole process was made automatic with increased throughput and lesser human involvement.” Less or lesser?
-
Page 3, Second paragraph change “in the already published model” to “published model”
-
Page 5, Case-I Assumption. There are two “4”
-
Page 6, “It was also used to for the removal of the completely filled cups from the exit point of the machine and placed them outside the system.” Need to rephrase “used to” or “use for”
-
Page 13, the paragraph “The benefits ….” The sentence is very wordy and I think you can make it more smoothly and concise.
10. Page 14. “Two equal parts each part equal to 45” I think there is missing an “and”
Reviewer 2 Report
The paper has fair features in terms of technical soundness, importance of results, clarity of presentation, awareness of the literature. The text is generally well organized and tidy, but some minor aspects need revision, The issues to clarify, correct and improve are listed below.
· A general revision of the text is suggested. Some repetitions in the text are present; they should be limited/avoided.
· Add the units of measure to the various variables in figures and tables.
· When possible, use min instead of minutes after a number.
· Introduction. This should be improved.
1) The goal of the paper should be presented with more clarity.
2) What most, clarify why the presented yogurt machine and the considered approach follows the principle of Industry 4.0 (this is shortly explained when citing Ref. [30].
3) The second part of introduction, when discussing the application on the yogurt machine, sounds a bit separated from the first part, which appears general, as many review papers on Industry 4.0 principles, paradigms, production planning, etc., are discussed.
4) Therefore, I suggest presenting other applications of industrial processes where the Industry 4.0 paradigms are exploited. For example:
· applications of process monitoring with appropriate strategies of data collection and cloud storage system (see A cloud-based monitoring system for performance assessment of industrial plants. Industrial & Engineering Chemistry Research, 59(6), pp.2341-2352.)
· application in the tannery industry: see An Automatic System for Modeling and Controlling Color Quality of Dyed Leathers in Tanneries IFAC-PapersOnLine, 54 (3), 164-169.
